

# Symmetry breaking of the cellular lobes closely relates to phylogenetic structure within green microalgae of the *Micrasterias* lineage (Zygnematophyceae)

Jiri Neustupa and  Jan Stastny

Department of Botany, Faculty of Science, Charles University Prague, Prague, Czech Republic

## ABSTRACT

Green microalgae of the *Micrasterias* lineage are unicellular microorganisms with modular morphology consisting of successively differentiated lobes. Due to their morphological diversity and peculiar morphogenesis, these species are important model systems for studies of cytomorphogenesis and cellular plasticity. Interestingly, the phylogenetic structure of the *Micrasterias* lineage and most other Desmidiales is poorly related to the traditional morphological characters used for delimitation of taxa. In this study, we focused on symmetry breaking between adjacent cellular lobes in relation to phylogeny of the studied species. While pronounced morphological asymmetry between the adjacent lobes is typical for some species, others have been characterized by the almost identical morphologies of these structures. We asked whether there is any detectable average shape asymmetry between the pairs of lobes and terminal lobules in 19 *Micrasterias* species representing all major clades of this desmidiacean lineage. Then, we evaluated whether the asymmetric patterns among species are phylogenetically structured. The analyses showed that the phylogeny was in fact strongly related to the patterns of morphological asymmetry between the adjacent cellular lobes. Thus, evolution of the asymmetric development between the adjacent lobes proved to be the key event differentiating cellular shape patterns of *Micrasterias*. Conversely, the phylogeny was only weakly related to asymmetry between the pairs of terminal lobules. The subsequent analyses of the phylogenetic morphological integration showed that individual hierarchical levels of cellular morphology were only weakly coordinated with regard to asymmetric variation among species. This finding indicates that evolutionary differentiation of morphogenetic processes leading to symmetry breaking may be relatively independent at different branching levels. Such modularity is probably the key to the evolvability of cellular shapes, leading to the extraordinary morphological diversity of these intriguing microalgae.

## INTRODUCTION

The green algal genus *Micrasterias* (Desmidiales, Zygnematophyceae) is typical by uniquely elaborate cellular shapes with multiple incisions dissecting the cells into multiple

Corresponding author
Jiri Neustupa,
neustupa@natur.cuni.cz

lobes and lobules that compose their biradially symmetric morphologies (*Prescott, Croasdale & Vinyard, 1977*; *Coesel & Meesters, 2007*). The arrangement and shapes of these morphological features are usually species specific; therefore, individual cells can be reliably identified by their cellular morphology. In contrast, clades of species as resolved by molecular phylogenies cannot be clearly defined by the shared morphological features (*Škaloud et al., 2011*).

Previous studies showed that the *Micrasterias* lineage of Desmidiales consists of at least eight monophyletic clades, including one comprising species of the morphologically divergent traditional genus *Triploceras* (*Hall et al., 2008*; *Gontcharov & Melkonian, 2011*; *Škaloud et al., 2011*). These clades do not unambiguously correspond to any of the formerly created infrageneric taxonomic units. Therefore, these clades have been informally assigned as A-H (*Škaloud et al., 2011*). In a prior study, *Neustupa & Stastny (2006)* illustrated that phenetic trees based on morphological and morphometric features of different *Micrasterias* species are considerably different from phylogenetic structure. Neither the shapes of the cellular polar lobes, emphasized in traditional taxonomy (*Prescott, Croasdale & Vinyard, 1977*; *Růžička, 1981*), nor the quotient of isoperimetric inequality that quantifies the deviation of the cellular shapes from circular outlines correlate with the molecular phylogeny of *Micrasterias* (*Škaloud et al., 2011*). In contrast, it has been shown that the degree of cellular branching is basically the only phenotypic feature with a significant phylogenetic signal across the *Micrasterias* lineage (*Škaloud et al., 2011*). It is known that this feature is closely related to the cell size and nuclear genome size of individual species (*Poulíčková et al., 2014*; *Neustupa, 2016*), which suggests that, unlike the previously studied shape features, the sizes of the cells are actually constrained by the phylogenetic history of these microalgae. However, the lack of size-unrelated characters correlated with the *Micrasterias* phylogeny implies that the principal morphogenetic patterns underlying the evolutionary diversification of this important model group for the study of cellular morphogenesis in eukaryotic microorganisms (*Kiermayer, 1970*; *Meindl, 1993*; *Lütz-Meindl, 2016*) remain unknown. Identification of phylogenetically conserved aspects of variation would indicate the evolutionary developmental patterns that drive the phenotypic evolution of this lineage with possibly the most complex cellular shapes among plants.

*Micrasterias* cells are bilaterally symmetric around multiple axes running through surface incisions; these separate the cell into cellular halves (semicells), semicels into lateral lobes and lateral lobes into successively branched lobules (Fig. 1A). The symmetry of these structures is set up in cellular morphogenetic processes during semicell development following the asexual mitotic cell division. Classical studies of *Kiermayer (1964)* and *Kiermayer (1981)* illustrated that the plasma membrane of the developing semicell contains the "membrane recognition areas" for the vesicles with the wall material formed in the Golgi apparatus and transported by actin microfilaments to actively growing areas (*Holzinger & Lütz-Meindl, 2001*; *Holzinger et al., 2002*). Apparently, the position of the membrane recognition sites is genetically fixed, which is demonstrated by species-specific differences in the symmetric morphologies of individual species of the lineage. *Kiermayer (1981)* showed that while the surfaces of the developing semicells cease to extend during the experimental turgor reduction, the primary wall material continues to accumulate

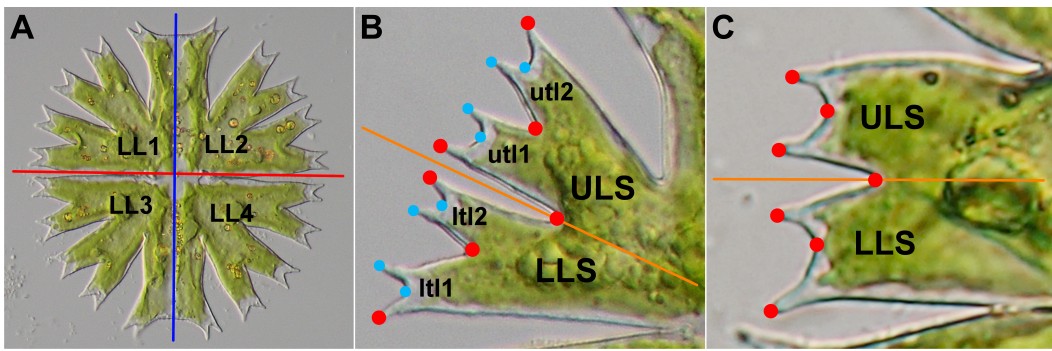

**Figure 1  Morphology of *Micrasterias* cells.** (A) The mature cell of *M. radians* var. *evoluta* with two semicells, each including two lateral lobes (LL1–LL4). Red line depicts the axis of symmetry between two opposite semicells. Blue line depicts the symmetry axis between two halves of each semicell. (B) The lateral lobe of *M. crux-melitensis* with the lower (LLS) and upper (ULS) lateral sublobes, each composed of two 3rd-order terminal lobules (ltl1, ltl2, utl1, utl2). The positions of seven landmarks depicting the sublobes are shown by red circles, and an additional eight landmarks depicting the terminal lobules are shown in ligth blue. The orange line depicts the axis of symmetry between two adjacent lateral sublobes. (C) The lateral lobe of *M. truncata* var. *pusilla* with two sublobes (LLS, ULS). The positions of seven landmarks depict the lateral sublobes. The orange line depicts the axis of symmetry between two adjacent lateral sublobes.

beneath the membrane, exactly reflecting the species-specific morphological pattern, i.e., the positions and sizes of individual lobes and incisions. In normally developing semicells, morphogenesis proceeds via a repeated spatiotemporal sequence consisting of discontinuation of primary wall growth at symmetrically arranged areas of the plasma membrane. These areas later become incisions, and the parts that continue to grow by tip elongation develop into lobules. Unlike in moss protonemata or in pollen tubes of vascular plants, tip growth of the lobules is multipolar, occurring simultaneously in different parts of a developing semicell (*Kiermayer & Meindl, 1989*; *Holloway & Harrison, 1999*; *Lütz-Meindl, 2016*). The genome-wide transcript profiling in *M. denticulata* showed that this tip growth phase is related to expression of expansin-like proteins similar to those involved in the cell morphogenesis of land plants (*Vannerum et al., 2011*).

However, in addition to multiple morphological symmetries, the cells of *Micrasterias* also typically include different levels of asymmetry. The development of opposite semicells is often separated by considerable time lags, implying that changes in environmental factors that affect morphogenetic processes, such as the external temperature (*Meindl, 1990*; *Neustupa, Stastny & Hodač, 2008*), may lead to pronounced morphological differences between two opposite mature semicells forming a single cell. Simultaneously, subtle but detectable asymmetry in shape was detected between two lateral lobes of the same semicell (*Savriama, Neustupa & Klingenberg, 2010*; *Neustupa, 2017*). This asymmetry is supposedly a result of morphogenetic noise during synchronous development of the semicell halves. It should be emphasized that both these asymmetric patterns of cellular morphology are typical by their side ambiguity (*Savriama, Neustupa & Klingenberg, 2010*; *Neustupa, 2013*) because, unlike bilateral animals or leaves of vascular plants, the cells of green algae typically lack any anterior-posterior differentiation (*McManus, Lewis & Schultz, 2011*;

*Lenarczyk & McManus, 2016*). Therefore, the left and right sides of the frontal views of cells cannot be assigned and analyzed among different individuals. Consequently, the analysis of asymmetric variations in the shapes of the semicells or their lateral lobes cannot separate the side-directed, average asymmetry between two structures of an entire population (i.e., directional asymmetry (DA)) from individual asymmetric deviations (i.e., fluctuating asymmetry (FA)).

However, complex morphology of *Micrasterias* cells includes additional structures within the lateral lobes that are symmetric to each other and are differentiated based on their relative positions. Each lateral lobe is typically split into two lateral sublobes, and in some species, these sublobes can be further divided into higher-order terminal lobules. These lateral sublobes within each lobe are bilaterally symmetric, and their side orientation can be assigned with respect to their position within the lateral lobe (Fig. 1). From the perspective of phenotype evolution in the *Micrasterias* lineage, the average asymmetry between adjacent pairs of the lateral sublobes in individual species may be of special interest. Similar to other repeated symmetric parts of organismal bodies, such as arthropod segments (*Savriama et al., 2017*), or cells in algal or bacterial filaments (*Graham et al., 2010*), symmetry of the different segments may be broken by different developmental signals specifying the position within the translational series. Thus, this idea may also be used for quantification and comparison of average shape differences in positional asymmetry, i.e., the symmetry breaking between adjacent sublobes within the *Micrasterias* cells. It should be noted that this component of the asymmetry between the adjacent sublobes basically corresponds to DA of the bilateral structures, such as animal body plans or vascular plant leaves (*Klingenberg, Barluenga & Meyer, 2002*). However, as it does not relate to any directionally determined left–right asymmetric differences, we refrain from using the explicit term DA for this positional asymmetry between the adjacent sublobes within the *Micrasterias* cells.

Interestingly, positional component of asymmetry between the sublobes is probably genetically fixed and typifies individual taxa. For example, the lateral sublobes of several *Micrasterias* species, such as *M. rotata*, *M. fimbriata*, and *M. apiculata*, are strongly asymmetric (*Coesel & Meesters, 2007*). Their lower lateral sublobe (LLS) is markedly compressed, and this apparent positional asymmetry has been considered one of the basic features used for their taxonomic definition (*Růžička, 1981*). On the other hand, there are also a number of taxa, such as *M. americana*, *M. furcata*, and *M. thomasiana*, with almost identical lateral sublobes (*Prescott, Croasdale & Vinyard, 1977*; *Růžička, 1981*). Thus, *Růžička (1981)* described the lateral sublobes of *M. thomasiana* as being "almost of the same width" and those of *M. americana* as being "approximately of the same width". These species may either lack any significant positional asymmetry in the shapes of the lateral sublobes, or it is subtle and is undetectable by qualitative microscopic observation. However, such positional asymmetry could possibly be identified by a morphometric analysis of the shape features of the adjacent sublobes in these *Micrasterias* species.

Studies of cellular morphogenesis have illustrated that the prominent positional asymmetry between the lateral sublobes in *M. rotata* is already established in the early phases of development in the asymmetric setting of the minimum accretion zones on

the plasma membrane of the developing semicell (*Kiermayer, 1981*; *Lütz-Meindl, 2016*). This leads to the development of a considerably narrower and less segmented LLS in comparison to its upper lateral sublobe (ULS) counterpart (Fig. 1). Detection of a similar pattern of positional asymmetry in shape of the adjacent lateral sublobes in other species of the *Micrasterias* lineage would imply that this feature is evolutionarily conserved and distributed throughout this phylogenetic lineage, and individual species differ only in the degree of this asymmetry. Conversely, it is possible that some species lack this type of asymmetry and that their average shapes of the LLSs and ULSs are not significantly different. This result would indicate that the splitting of two membrane parts, which precedes the development of the lateral sublobes, is truly dichotomic. This pattern was largely assumed by the mathematical models of cellular morphogenesis in *Micrasterias* based on the two-morphogen reaction–diffusion activity, controlling the growth and branching of the developing membrane parts within the growing semicell (*Holloway & Harrison, 2008*; *Holloway, 2010*). Finally, the analysis might show that significant positional asymmetry is present in most or all of the studied species but that the patterns of this asymmetry are markedly different among taxa. Thus, in this scenario, some species would not have the relatively more compressed LLSs, typical for *M. rotata* or *M. fimbriata*, but would possibly exhibit a different and significant pattern of positional asymmetry in the shape of the lateral sublobes. This result would then lead to the question of how these different asymmetric patterns are related to the evolutionary history of *Micrasterias*. By evaluation of these scenarios, we would like to illustrate the way in which the cellular morphogenetic mechanisms leading to fixed asymmetry of individual structures within the species-specific morphology may be related to evolutionary history of the lineage and how this phenomenon can be reflected in the observed morphological diversity of these unicellular organisms.

The lateral lobes of many *Micrasterias* species are successively differentiated into higher-order lobules (Fig. 2). Recently, it was shown that shape variation of the adjacent terminal lobules in *M. compereana* is tightly integrated (*Neustupa, 2017*). However, such high covariation may not be present in the asymmetric variation of these lobules across species. If the patterns of the among-species asymmetric variation prove to be tightly integrated among different levels of cellular differentiation, then the morphogenetic mechanisms of symmetry breaking likely act relatively uniformly during successive development of the lobes and their subordinate sublobes and lobules. In species with pronounced positional asymmetry between the lateral sublobes, such as *M. rotata,* this scenario would imply that similar asymmetric pattern is shared by adjacent pairs of sublobes. Likewise, species with relatively equal lateral sublobes would also exhibit almost identical pairs of adjacent terminal lobules.

Alternatively, we may find that the dynamics of asymmetry among different levels of cellular branching is unrelated among the species. This outcome would suggest that individual stages of the morphogenetic process are relatively independent and evolved separately in different species. This result would indicate minimal evolutionary coordination among the successive stages of cellular development, which may facilitate the

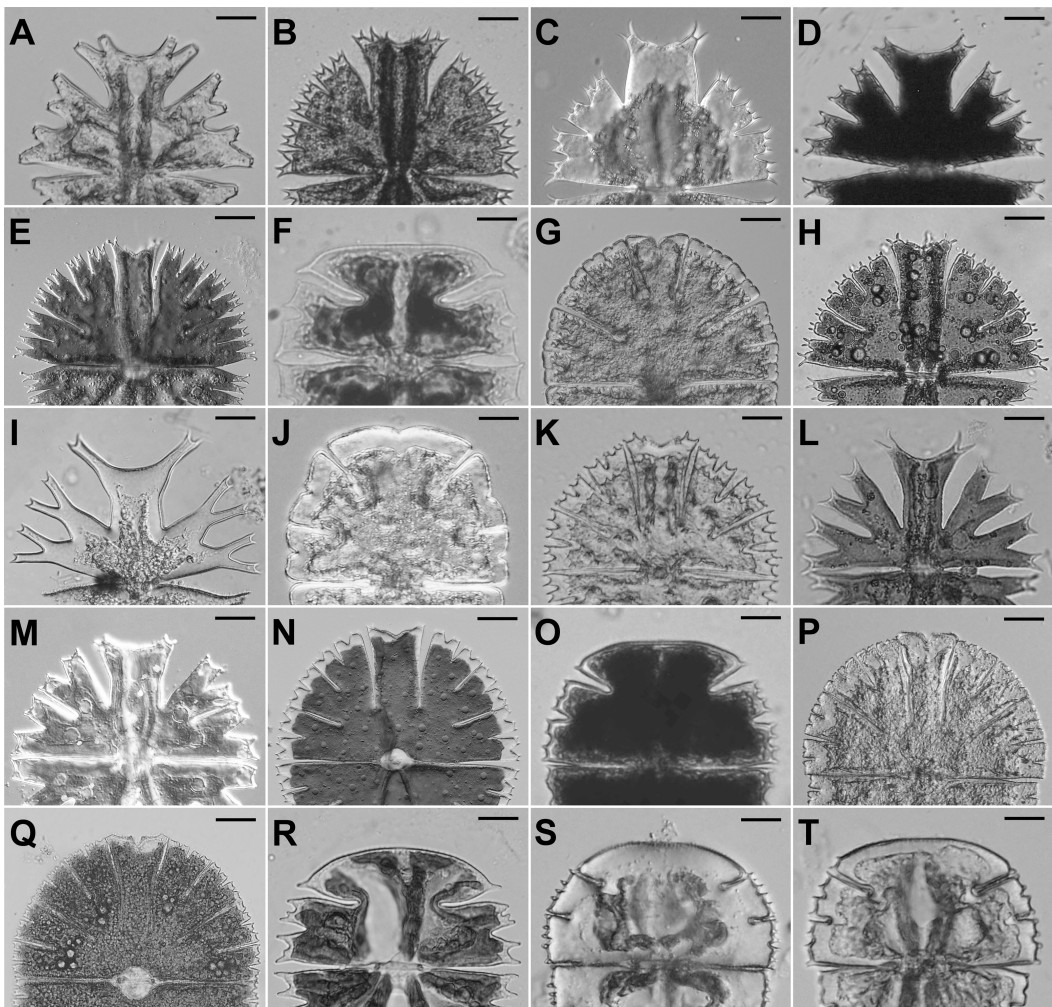

**Figure 2** **Semicells of 19 *Micrasterias* taxa.** (A) *M. americana*. (B) *M. apiculata*. (C) *M. brachyptera*. (D) *M. crux-melitensis*. (E) *M. compereana*. (F) *M. decemdentata*. (G) *M. denticulata*. (H) *M. fimbriata*. (I) *M. furcata*. (J) *M. jenneri*. (K) *M. papillifera*. (L) *M. radians* var. *bogoriensis*. (M) *M. radians* var. *evoluta*. (N) *M. rotata*. (O) *M. semiradiata*. (P) *M. thomasiana*. (Q) *M. thomasiana*. (R) *M. truncata* var. *pusilla*. (S) *M. truncata* var. *quadrata*. (T) *M. truncata* var. *truncata*. Scale bars = 12 μm (F, R–T), 20 μm (D, J, K, O), 25 μm (A, I, L, M), and 30 μm (B, C, E, G, H, N, P, Q).

evolution of different shape patterns at individual stages and may significantly contribute to the observed morphological diversity of these organisms.

In summary, the present study evaluates the following specific questions:

(1) What are the patterns of positional asymmetry in the shapes of the lateral sublobes of *Micrasterias* species? Are these asymmetric patterns related to the phylogenetic relationships among the species?

(2) Is the species-level positional asymmetry in the shapes of the lateral sublobes also shared by the pairs of 2nd-order sublobes within the ULS and LLS, indicating homogenous morphogenetic mechanisms of symmetry breaking across successive levels of cellular

development? In other words, is there a significant phylogenetic morphological covariation in asymmetric variation between different levels of cellular lobes?

These questions were answered by performing geometric morphometric analyses of the cellular shapes of 19 *Micrasterias* species. The answers to these questions should help us describe the evolutionary diversification of morphogenetic patterns, which lead to specific morphologies of individual taxa and are reflected in the observed diversity of the extant taxa.

## MATERIAL & METHODS

### Sampling, cultivation and data acquisition

The data were based on populations of 19 *Micrasterias* taxa (Fig. 2, Table S1) representing all major phylogenetic clades of this desmidiacean lineage as previously recovered by *Škaloud et al. (2011)*. The samples from the natural communities were immediately fixed by using Lugol's solution and stored in the dark at 17 °C. The strains were cultured in MES-buffered DY IV liquid medium at 23 °C and illuminated at 40 $\mu$mol photons m$^{-2}$ s$^{-1}$ with LED EVOLVEO natural light 9 W bulbs under a light:dark (L:D) regime of 12:12 h. Notably, 5 of the studied taxa have been treated as subspecies in traditional taxonomic monographs (*Růžička, 1981*). However, published phylogenetic data clearly illustrated that these taxa form well-delimited species-level evolutionary units within the genus *Micrasterias* (*Nemjová et al., 2011*; *Škaloud et al., 2011*; *Neustupa, Stastny & Škaloud, 2014*). Therefore, in our analses we treated these taxa as separate units.

For each taxon, a total of 75 mature semicells were photographed. Microphotographs were taken on an Olympus BX51 light microscope (Olymous, Shinjuku, Japan) with a Bresser MikroCam SP 5.0 digital camera (Bresser, Rhede, Germany). The left lateral lobes of each semicell were used for digitization of landmarks and subsequent analyses. In total, seven landmarks (Figs. 1B–1C) were depicted on all the specimens using TpsDig software, ver. 2.15 (*Rohlf, 2015*). In 14 species that exhibited lobe differentiation into 3rd-order lobules, an additional eight landmarks were digitized (Fig. 1B), yielding configurations consisting of 15 landmarks. To assess the measurement error in the analyses of morphological symmetry and asymmetry, all landmarks were digitized twice. In the first digitization, the landmarks were registered from the lower margin of the LLS. In contrast, the landmarks in the second digitization were captured counterclockwise from the upper ULS margin. Subsequently, the landmarks from the second digitization were relabeled to match the labels of the first digitization (*Neustupa, 2017*).

### Procrustes ANOVA

Four parallel configurations were used for the analysis of object symmetry in individual taxa using a series of Procrustes ANOVA tests. Two lateral sublobes, the LLS and ULS, were compared in two analyses (Fig. 1). The first comprised configurations of seven landmarks in all 19 taxa. Then, all 15 landmarks were used for this analysis in 14 taxa with semicells differentiated into 3rd-order lobules (Fig. 1B). In addition, in these 14 taxa, the object symmetry between the opposite terminal lobules within both the LLS (ltl1 vs. ltl2) and ULS (utl1 vs. utl2) were also evaluated using the configurations of the seven landmarks depicted

on either of these sublobes (Fig. 1B). Thus, Procrustes ANOVA of the object symmetry between two lobules of the LLS was investigated using configurations of landmarks no. 1–7 and the object symmetry within the ULS with landmarks no. 9–15 (Fig. 1B).

In all these parallel datasets, the geometric morphometric analysis was based on the generalized Procrustes analysis (GPA) of the two-dimensional landmark coordinates (*Zelditch, Swiderski & Sheets, 2012*). The resulting Procrustes coordinates were used in the multivariate nonparametric Procrustes ANOVA models to evaluate symmetric variation among individuals, the asymmetric components of positional asymmetry (equivalent to DA) and FA, and the digitization error (*Klingenberg, Barluenga & Meyer, 2002*; *Klingenberg, 2015*). The analysis was based on a matrix of tangent Procrustes distances among the original and reflected/relabeled configurations of landmarks. Observed variation was partitioned into several sources, such as the shape differences among the individuals (i.e., different cells); average shape differences between the upper and lower sublobes within cells, signifying positional asymmetry; the interaction term between these two main effects, denoting FA; and the measurement error, which illustrated the amount of variation spanned by digitization imprecision. The significance of individual effects was evaluated by randomization tests comparing the observed $F$-values to those obtained by 999 random permutations of the matrix rows comprising the shape data of the specimens (*Klingenberg, Barluenga & Meyer, 2002*).

These analyses were conducted using the functions *gpagen* and *bilat.symmetry* implemented in the *geomorph* package, version 3.0.5 (*Adams & Otárola-Castillo, 2013*), of R, version 3.2.3 (*R Development Core Team, 2016*).

## Principal component analysis (PCA) and structure of positional asymmetry

While the present study is focused on the patterns of average asymmetry between the lateral lobes and lobules among individual taxa, the overall shape features of these lateral lobes are also considerably different (*Neustupa & Stastny, 2006*; *Škaloud et al., 2011*, Fig. 2). These differences can be expressed as the symmetric variation in the shape of the configurations among the species. Thus, the analysis separated the symmetric variation from the components of bilateral asymmetry. Then, we focused on the asymmetric variation among the configurations depicting the average positional asymmetry of individual taxa. These configurations were yielded by the Procrustes ANOVA models together with their mirrored versions. Therefore, these pairs of asymmetric configurations were obtained for each of the studied taxa. The configurations were merged into a single dataset and subjected to GPA. The Procrustes-aligned coordinates of these original and mirrored average asymmetrical configurations were then subjected to PCA. Resulting principal components (PCs) described either purely symmetric or asymmetric variation with regard to the axis of symmetry dividing the sublobes (*Savriama, Neustupa & Klingenberg, 2010*; *Savriama & Klingenberg, 2011*). Thus, the asymmetric PCs described the purely asymmetric shape changes with respect to the axis of bilateral symmetry, showing the patterns of the shape changes in average bilateral asymmetry among the studied taxa. The ideally symmetric configuration with identical sublobes was placed at the center of the

ordination space, and the original and mirrored configurations of each specimen were placed at opposite positions on each PC. Differences in any particular configuration from the center of this ordination space explicitly represent the degree and direction of positional asymmetry typical for a given species. In parallel, PCA of the purely asymmetric variation of original configurations, without their mirrored copies, yielded PCs centered at the average asymmetric configuration of all studied taxa. This morphospace was used for mapping the patterns of asymmetry among the taxa onto the phylogeny of the *Micrasterias* lineage.

PCAs of the aligned configurations were conducted using the function *procGPA* in the *shapes* package, ver. 1.1-13 (*Dryden, 2016*), of R (*R Development Core Team, 2016*) and in TpsRelw, version 1.42 (*Rohlf, 2015*). This software was also used for reconstruction of the transformation grids depicting shape changes spanned by individual PCs.

## Phylogenetic analysis and morphological integration

The alignment of the 18S rDNA sequences created by *Neustupa (2016)* consisted of 1641 nucleotides with 116 variable positions. Two separate alignments were used. The first dataset, consisting of 19 *Micrasterias* species, was related to the morphometric analysis based on seven landmarks characterizing the shape of the sublobes. The second dataset included 14 species with sublobes differentiated into 3rd-order lobules that were used for the morphometric analyses of the bilateral symmetry between the LLS and ULS based on 15 landmarks as well as for the analyses within these sublobes. The alignments are available at https://doi.org/10.5063/F1GF0RQS.

The optimal evolutionary model for the maximum likelihood (ML) analyses was chosen on the basis of the Bayesian information criterion (BIC) implemented by the *modeltest* function of the *phangorn* package, ver. 1. 7-4 (*Schliep, 2011*), in R (*R Core Team, 2015*). The GTR-G-I model reached the lowest BIC value in both alignments, and consequently, this model was selected for the ML phylogenetic analyses, which were conducted using the functions *pml* and *optim.pml* implemented in the *phangorn* package. Bootstrap supports of the nodes of the phylogenetic trees were calculated by nonparametric bootstrap analysis based on 999 replicates using the *bootstrap.pml* function of the *phangorn* package.

Two parallel strategies for finding topologies with the lowest parsimony score were used in the maximum parsimony (MP) analyses of both datasets. The *optim.parsimony* function was used for the nearest-neighbor tree rearrangements, and *pratchet* was used for the parsimony ratchet searches (*Nixon, 1999*). Bootstrap analyses of the nodes of the optimal MP phylogenetic tree were based on 999 replicates. Phylogenetic trees were graphically adjusted in FigTree, ver. 1.3.1 (*Rambaut, 2009*).

Phylogeny was mapped onto the structure of the shape asymmetry by squared-change parsimony in MorphoJ, ver. 1.06d (*Klingenberg, 2011*). Then, the evolutionary trajectories were visualized in the shape spaces of individual PCAs. The phylogenetic signal in these shape spaces was evaluated by permutation tests simulating the null hypothesis of the absence of any relation to phylogenetic structure by randomly permuting the shape data among the terminal nodes of the trees and computing the total amount of squared change summed over all branches (*Klingenberg & Gidaszewski, 2010*). The tests were based on 9,999 random permutations.

Phylogenetic morphological integration between the asymmetric variation at different levels of cellular branching was estimated using the two-block partial least squares (PLS) analysis that also accounted for the phylogeny of the taxa under the Brownian motion model of evolution (*Adams & Felice, 2014*).

The two-block PLS quantifies covariation between two Procrustes-aligned morphometric datasets by singular value decomposition of the covariance matrix (*Zelditch, Swiderski & Sheets, 2012*). The resulting axes, which have also been called singular warps, successively describe the shape variation in one dataset with the highest covariation with the second dataset (*Bookstein et al., 2003*). Linear correlation between the first pair of singular warps has been called the PLS correlation ($r_{PLS}$) and has been widely used as a measure of morphological integration between two morphological structures (*Klingenberg, 2014*). In a phylogenetic context, the evolutionary change is modeled by a Brownian motion and is described by an evolutionary covariance matrix yielded by phylogenetic generalized least squares (*Adams & Felice, 2014*). The significance of the $r_{PLS}$ values between the evolutionary PLS scores from two blocks of the phylogenetically corrected datasets is assessed by comparison to a random distribution of values produced by permutation of species on the tips of the phylogeny in the first dataset in relation to the second dataset. This random distribution was based on 999 replicates.

In the present study, phylogenetic morphological integration was evaluated in 14 species with lobes differentiated in the 3rd-order lobules. In this dataset, we compared the covariation between the pairs of configurations spanning the asymmetric variation among the species means. Three sets of configurations were considered. The first set consisted of the entire dataset of 15 landmarks describing bilateral asymmetry between the LLS and ULS. Then, the landmarks no. 1–7 and 9–15 formed two datasets that described bilateral asymmetry between two terminal lobules of the LLS and two lobules of the ULS, respectively (Fig. 1B). Thus, three parallel morphological integration analyses were conducted between the pairs of these configurations. The analysis utilized the ML tree produced by the phylogenetic analysis and was conducted using the *phylo.integration* function implemented in the *geomorph* package (*Adams & Otárola-Castillo, 2013*).

## RESULTS

Two lateral sublobes, namely, the LLS and ULS, were significantly asymmetric in all 19 studied species (Table 1; Table S2). The observed positional asymmetry reached extreme values in *M. rotata*, *M. compereana*, and *M. fimbriata*. In these taxa, positional asymmetry represented the single most important component of the shape variation in the Procrustes ANOVA models, spanning more than 80% of the total variation between the sublobes (Table 1). This asymmetric pattern largely consisted of a considerably compressed LLS (Figs. 3A–3B), which is a well-known morphological feature of these taxa. In addition to these species, LLS was strongly compressed in *M. jenneri*, *M. apiculata* and *M. papillifera*, which was indicated by the negative scores of these taxa on PC1, which encompassed most of the variation in the PCA of the asymmetric shape variation among the original and mirrored mean configurations of the species (Figs. 3A–3B). In most remaining species, the amounts

**Table 1** **The mean squares (MS), percentages of variance ($R^2$) and $p$-values of positional asymmetry in individual Procrustes ANOVA analyses of bilateral symmetry.** The abbreviations identify structures depicted in Fig. 1.

| Species | LLS vs. ULS (7 LM) | | | LLS vs. ULS (15 LM) | | | ltl1 vs. ltl2 (7 LM) | | | utl1 vs. utl2 (7 LM) | | |
|---|---|---|---|---|---|---|---|---|---|---|---|---|
| | MS | $R^2$ | $p$ | MS | $R^2$ | $p$ | MS | $R^2$ | $p$ | MS | $R^2$ | $p$ |
| *M. americana* | 0.044 | 0.044 | 0.001 | X | X | X | X | X | X | X | X | X |
| *M. apiculata* | 0.745 | 0.638 | 0.001 | 0.532 | 0.574 | 0.001 | 0.705 | 0.469 | 0.001 | 0.158 | 0.178 | 0.001 |
| *M. brachyptera* | 0.189 | 0.122 | 0.001 | 0.131 | 0.099 | 0.001 | 0.113 | 0.040 | 0.001 | 0.540 | 0.182 | 0.001 |
| *M. crux-melitensis* | 0.043 | 0.056 | 0.001 | 0.055 | 0.075 | 0.001 | 2.029 | 0.637 | 0.001 | 0.562 | 0.289 | 0.001 |
| *M. compereana* | 2.959 | 0.848 | 0.001 | 1.973 | 0.806 | 0.001 | 0.113 | 0.087 | 0.001 | 0.120 | 0.132 | 0.001 |
| *M. decemdentata* | 0.881 | 0.440 | 0.001 | X | X | X | X | X | X | X | X | X |
| *M. denticulata* | 0.528 | 0.497 | 0.001 | 0.355 | 0.445 | 0.001 | 0.721 | 0.480 | 0.001 | 0.246 | 0.260 | 0.001 |
| *M. fimbriata* | 3.406 | 0.839 | 0.001 | 2.439 | 0.811 | 0.001 | 0.956 | 0.396 | 0.001 | 0.240 | 0.227 | 0.001 |
| *M. furcata* | 0.077 | 0.049 | 0.001 | 0.068 | 0.046 | 0.001 | 0.027 | 0.015 | 0.002 | 0.007 | 0.004 | 0.146 |
| *M. jenneri* | 1.703 | 0.761 | 0.001 | X | X | X | X | X | X | X | X | X |
| *M. papillifera* | 0.606 | 0.511 | 0.001 | 0.445 | 0.442 | 0.001 | 0.069 | 0.054 | 0.001 | 0.350 | 0.255 | 0.001 |
| *M. radians* var. *bogoriensis* | 0.023 | 0.015 | 0.001 | 0.037 | 0.028 | 0.001 | 0.084 | 0.032 | 0.001 | 0.188 | 0.061 | 0.001 |
| *M. radians* var. *evoluta* | 0.036 | 0.023 | 0.001 | 0.031 | 0.024 | 0.001 | 0.085 | 0.038 | 0.001 | 0.003 | 0.001 | 0.469 |
| *M. rotata* | 4.303 | 0.878 | 0.001 | 3.017 | 0.828 | 0.001 | 0.022 | 0.016 | 0.001 | 0.675 | 0.379 | 0.001 |
| *M. semiradiata* | 0.600 | 0.406 | 0.001 | 0.387 | 0.321 | 0.001 | 0.402 | 0.195 | 0.001 | 0.286 | 0.103 | 0.001 |
| *M. thomasiana* | 0.252 | 0.421 | 0.001 | 0.176 | 0.376 | 0.001 | 0.373 | 0.416 | 0.001 | 0.025 | 0.052 | 0.001 |
| *M. truncata* var. *pusilla* | 1.582 | 0.448 | 0.001 | X | X | X | X | X | X | X | X | X |
| *M. truncata* var. *quadrata* | 0.140 | 0.224 | 0.001 | X | X | X | X | X | X | X | X | X |
| *M. truncata* var. *truncata* | 0.182 | 0.228 | 0.001 | 0.125 | 0.157 | 0.001 | 0.562 | 0.295 | 0.001 | 0.070 | 0.039 | 0.001 |

of positional asymmetry were considerably lower, but the MS for positional asymmetry still proved to be at least nine times higher than the MS for FA even in species positioned close to the ideally symmetric midpoint on PC1, such as *M. furcata*, *M. radians* var. *evoluta*, and *M. crux-melitensis* (Figs. 3A–3B; Table S2). In *M. semiradiata*, *M. decemdentata* and *M. truncata* var. *pusilla*, which occupied the positive parts of the morphospace along the PC1, the pattern of average asymmetry between the lateral sublobes was typical by a significantly greater compression of the ULS than that of the LLS (Fig. 3A).

The average asymmetry between the two terminal lobules of the LLS was also consistently significant in all 14 taxa possessing 3rd-order lobules (Table 1; Table S2). Interestingly, asymmetry between adjacent terminal lobules was generally subtle in taxa with extreme positional asymmetry between the lateral sublobes, such as *M. rotata* and *M. compareana* (Table 1). On the other hand, several other species, such as *M. crux-melitensis* and *M. denticulata*, had pronounced positional asymmetry between the terminal lobules within the LLS or ULS, although their lateral sublobes were much less asymmetric (Table 1; Fig. 3). In *M. crux-melitensis* and, to a lesser extent, in *M. fimbriata* and several other taxa, the lower-most terminal lobule was considerably more expanded than its adjacent counterpart. In contrast, *M. denticulata* and *M. thomasiana* were typical by distinctly opposite patterns of asymmetry between these lobules (Fig. 3C).

The average shape asymmetry between the terminal lobules of the ULS was highly significant in 12 out of the 14 analyzed taxa. In two species, *M. furcata* and *M. radians*

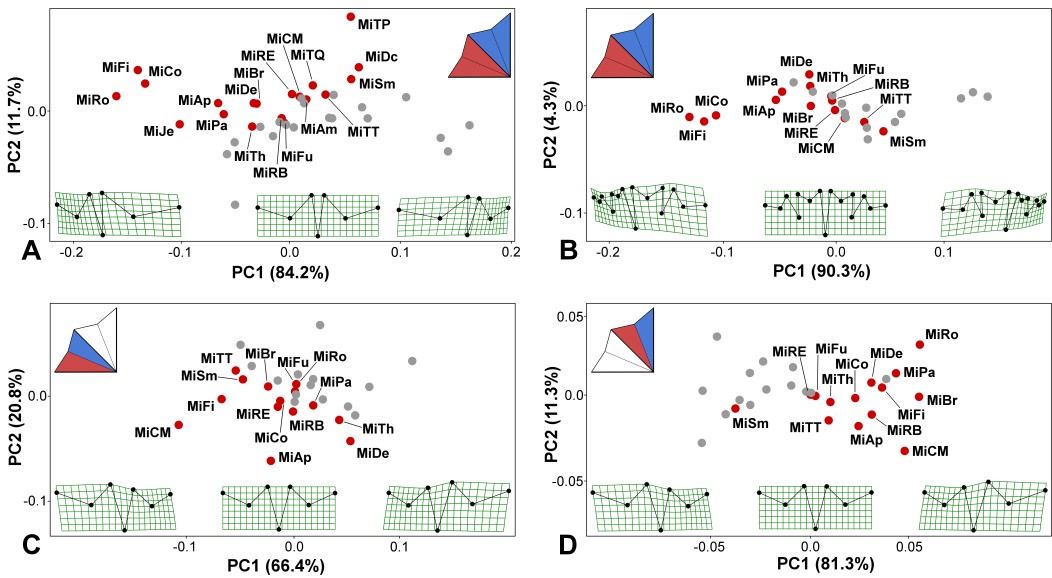

**Figure 3** **PCA ordination plots showing the differences in positional asymmetry of the adjacent lateral sublobes.** The red dots depict the position of the original asymmetric configurations for each species. The gray dots show their mirrored copies. Deformation grids illustrate the ideally symmetric configuration (located at the midpoint of each plot) and the configurations typical for 3 × S.D. positions on both extremes of PC1. The schemes in the upper left corners show the types of bilateral asymmetry illustrated by each particular analysis. Abbreviations of taxa correspond to those in Table S1. (A) Positional asymmetry between the LLS and ULS based on configurations of seven landmarks. (B) Positional asymmetry between the LLS and ULS based on configurations of 15 landmarks. (C) Positional asymmetry patterns between the two terminal lobules within the LLS. (D) Positional asymmetry patterns between the two terminal lobules within the ULS.

var. *evoluta*, the average shapes of these lobules did not differ compared to individual asymmetric deviations (Table 1; Table S2). On the other hand, this type of positional asymmetry was highly pronounced in *M. brachyptera*, a species with rather modest levels of average asymmetry between both the lateral sublobes and the terminal lobules of the LLS. Together with *M. rotata* and *M. crux-melitensis*, this species was characteristic by positional asymmetry typified by a significantly large expansion of the upper-most terminal lobule (Fig. 3D). The opposite pattern was only typical for *M. semiradiata*, which was the only species occupying the negative parts of PC1 in the morphospace illustrating the asymmetric variation between the terminal lobules of ULS (Fig. 3D).

The ML and MP phylogenetic analyses yielded topologies that were very similar to the previously reported multigenic trees of the *Micrasterias* lineage (*Škaloud et al., 2011*; *Neustupa, Stastny & Škaloud, 2014*). The firmly supported clade "A", including the species complex of *M. truncata, M. furcata* (the type species of the genus) and several other taxa, proved to be in a sister position to the remaining clades (Fig. S1). The closely related species *M. rotata* and *M. fimbriata* formed a joint lineage with *M. brachyptera* and *M. compereana*, corresponding to clade "B" *sensu Škaloud et al. (2011)*. In addition, *M. denticulata, M. jenneri* and *M. thomasiana* also formed a tightly supported clade overlapping with clade "D" of the aforementioned phylogenetic study.

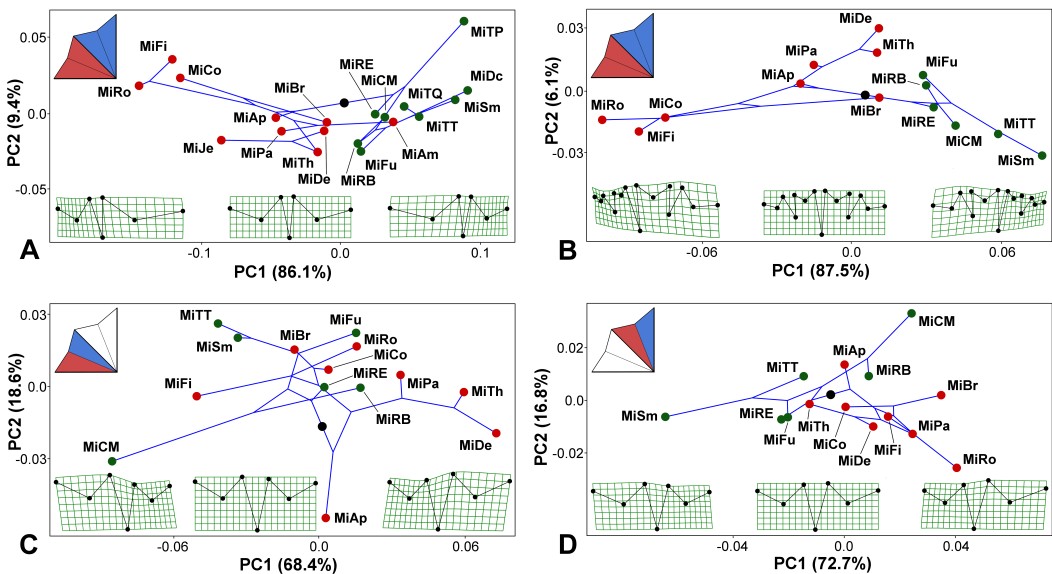

**Figure 4** *Micrasterias* **phylogeny mapped onto the morphospaces represented by the PC1 vs. PC2 ordination plots based on the patterns of positional asymmetry within the species.** Deformation grids illustrate the configuration of the asymmetric grand mean and those typical for 3 × S.D. positions on both extremes of PC1. The schemes in the upper left corners show the types of bilateral asymmetry illustrated by each particular analysis. Abbreviations of taxa correspond to those in Table S1. Members of the monophyletic clade "A", firmly supported in the phylogenetic analysis, are depicted by green circles. (A) PCA based on the positional asymmetry between the LLS and ULS registered by the configurations of seven landmarks. (B) PCA based on the positional asymmetry between the LLS and ULS registered by the configurations of 15 landmarks. (C) PCA of the positional asymmetry patterns between the two terminal lobules within the LLS. (D) PCA of the positional asymmetry patterns between the two terminal lobules within the ULS.

Mapping the phylogeny onto the morphospaces showed a strong phylogenetic signal in the among-species patterns of bilateral asymmetry between the LLS and ULS (Figs. 4A–4B). The null hypothesis presuming the lack of any phylogenetic signal in both these morphospaces was rejected by the permutation tests with $p = 0.0003$ in the 7-landmark dataset and $p = 0.0001$ in the dataset based on 15 landmarks. The phylogenetic signal was largely based on the difference in positional asymmetry between the members of clade "A" and the remaining taxa of the lineage. All the clade "A" taxa had either very similar shapes of ULS and LLS, with only slight positional asymmetry between these sublobes, or their ULS was considerably more compressed than their LLS (Figs. 4A–4B). Conversely, members of most other *Micrasterias* clades showed the opposite pattern of this asymmetry, with significantly compressed LLSs. However, there was a single prominent exception to this scenario. *M. americana*, which clearly clusters outside of clade "A" in the phylogenetic tree, was positioned close to the members of this clade in the right part of the analyzed morphospace (Fig. 4A). Looking at the morphospace of the original and mirrored configurations, we can see that the LLS of this species was slightly but significantly more expanded than the ULS (Fig. 3A).

The morphospaces describing the patterns of the bilateral asymmetric variation between the adjacent terminal lobules within the LLS and ULS proved to be weakly related to the phylogenetic structure of the lineage (Figs. 4C–4D). The null hypothesis was rejected with rather modest values of $p = 0.0310$ for the lobules within the LLS and $p = 0.0140$ for the ULS.

Phylogenetic morphological integration in the among-species asymmetric variation of the terminal lobules within both lateral sublobes was weak and insignificant. The PLS analysis evaluating the degree of covariation while accounting for phylogeny yielded $r_{PLS} = 0.639$ with $p = 0.087$. Thus, this analysis did not reject the null hypothesis of total independence between the average asymmetric patterns of the lobules within the LLS and ULS among the analyzed species when their phylogenetic relationships were taken into consideration. Likewise, the average asymmetry between the LLS and ULS was not significantly covariated with the asymmetry of the terminal lobules within the LLS ($r_{PLS} = 0.633$, $p = 0.121$). Finally, the phylogenetic morphological integration in the asymmetric patterns between the ULS and LLS and those between the lobules within the ULS was weakly significant, with $r_{PLS} = 0.744$ and $p = 0.021$.

## DISCUSSION

Our analyses showed that positional asymmetry is invariably present in the shape of the bilaterally symmetric lateral sublobes of *Micrasterias* cells, including those taxa that have traditionally been characterized as possessing identical lateral sublobes (*Prescott, Croasdale & Vinyard, 1977*; *Růžička, 1981*). It has also been shown that this asymmetry is strongly related to phylogenetic history of the lineage. Most infrageneric clades shared the pattern of the LLS being more compressed than the ULS, which was most pronounced in the species belonging to clade "C", such as *M. rotata* and *M. fimbriata*. However, two infrageneric clades were distinctly different. The species-rich clade "A" had either very subtle shape differences between LLS and ULS or there was a pronounced positional asymmetry in the opposite manner to that observed in the rest of the genus. In particular, the subclade that included *M. decemdentata*, *M. semiradiata*, and the members of the *M. truncata* species complex was characterized by considerably greater expansion of the LLS than that of the ULS. In addition, there was the peculiar case of *M. americana*, a species that is unrelated to members of clade "A" and forms a clade of its own within the genus, which also shared this pattern of positional asymmetry. It has been shown that *M. americana* is closely phylogenetically related to the non-European *M. mahabuleswarensis, M. hardyi* and *M. muricata* (*Teiling, 1956*; *Škaloud et al., 2011*). Therefore, it would be interesting to ascertain whether the asymmetry between the sublobes in these taxa is similar to that of *M. americana*, which would indicate that this may be a feature of this entire group, very similar to the scenario observed in members of clade "A". It should be mentioned that the polar lobes in members of clade "H" are typical by distinct deviations from the front view plane. In *M. muricata*, this pattern of distinctly three dimensional symmetry involves the lateral lobes, too (*Holzinger, 2000*; *Harrison, 2011*). Thus, this indicates that morphogenetic patterning in this clade differs from most other *Micrasterias* species, including the species clustering into clade "A".

It is unclear which type of positional asymmetry between the LLS and ULS should be considered the ancestral character state within the lineage as a whole. The comprehensive phylogeny of Desmidiaceae, based on the rbcL gene sequences, indicated that clade "A" might be a sister lineage to all other members of the genus (*Gontcharov & Melkonian, 2011*), which was also shown by our 18S rDNA phylogenetic analysis. Conversely, the multigene phylogeny described by *Škaloud et al. (2011)* recovered this clade as a crown lineage, favoring the scenario of the compressed LLS, which is present in most other clades, as a plesiomorphic character. However, weak statistical support for the topology of the deep branches of the *Micrasterias* phylogeny does not allow any definite conclusions regarding this matter.

In any case, the present study has shown that asymmetry between the lateral sublobes of cells constrains the morphological evolution of individual species-level taxa within the clades. In contrast to most other morphological characteristics, the positional asymmetry between LLS and ULS cannot be easily altered during microevolutionary differentiation of these microalgae. This asymmetry has been fixed relatively deep in the evolutionary history of each clade and is shared by all of the members of the clade despite their considerable morphological diversity. In terms of cellular morphogenesis, this result indicates that the position of the minimum accretion zones on developing lateral lobes, which determines the location of the incision between both lateral sublobes (*Kiermayer, 1981*; *Meindl, 1993*), is evolutionarily fixed at a level exceeding the boundaries among individual species.

Interestingly, the trends in the among-species asymmetric variation between the LLS and ULS were not corroborated by similar analyses at the level of adjacent terminal lobules within the lateral sublobes. The asymmetry in the shapes of the lobules was significant in most taxa, but the patterns of this asymmetry were only marginally related to the phylogenetic structure. Likewise, this among-species asymmetric variation was only weakly integrated with that at the levels of the entire lateral lobes. In addition, phylogenetic morphological integration in asymmetry between the lobules within the LLS and ULS was also insignificant. This result suggests that the asymmetric patterns at the individual hierarchic levels of cellular branching are almost independent. In other words, individual species may have their own type of evolutionarily fixed positional asymmetry at different levels of bilaterally symmetric structures formed by pairs of the lateral sublobes or the subordinate lobules. These conclusions generally concur with the characteristics of the theoretical reaction–diffusion models of *Micrasterias* cellular development (*Holloway & Harrison, 1999*). These models involve multiple independent centers of tip growth and cell wall patterning, leading to the development of individual sublobes and lobules (*Holloway, 2010*). Thus, according to the reaction–diffusion models, the growth patterns of cells belonging to different species should be weakly coordinated among different hierarchical levels, which was largely confirmed by the present study. However, it should be noted that these results relate to among-species comparisons of integration. While phylogenetic patterns of average asymmetry among different structures within lateral lobes are relatively independent, shape variation of adjacent lobules within a single population is tightly integrated (*Neustupa, 2017*).
Although it has now been demonstrated that species-level average asymmetry among different lobules is relatively independent, it should be noted that most of the morphological differences among the species are described by their symmetric differences. Therefore, future studies of morphological integration among different parts of *Micrasterias* cells in the phylogenetic context should also consider the symmetric differences among the configurations typical for individual species. Detection of a significant phylogenetic modularity among the lobes and lobules of different taxa would provide important additional knowledge regarding the phenotypic evolution of *Micrasterias*. Modularity, typified by minimal developmental coordination (i.e., minimal integration) among the parts of a structure that belong to different evolutionary modules (*Klingenberg, 2008*), has been known to promote phenotypic diversification among taxa via differential evolutionary rates of morphological change across units (*West-Eberhard, 2003*; *Larouche, Zelditch & Cloutier, 2018*). Thus, evolutionary independence among successive lobes and lobules of *Micrasterias* would be a strong indication that such phylogenetic modularity is a prerequisite for the evolvability of their cellular shapes. This might have been the key feature of Desmidiales leading to their extraordinary morphological diversity, coupled with taxonomic differentiation of more than 2,300 extant species and successful colonization of a wide array of freshwater microhabitats. Data on modularity of unicellular organisms are still very scarce but, in the light or our results, it will of special interest to look after the modularity of morphologically complex cells in other microalgal groups, such as Chlorophyceae and the diatoms, to find out if modular arrangement of cells is actually a more general prerequisite for prolific radiation in unicellular eukaryotes.

*Micrasterias* species proved to be intriguing and useful model organisms for the investigation of quantitative morphology and phenotypic plasticity at the cellular level. The flat cells of these species, with straightforward 2D representation of morphological variation and hierarchical differentiation of cellular lobes, provide multiple possibilities for evaluating different scenarios of cellular shape diversification, symmetry and asymmetry, or morphological integration among individual cellular parts (*Neustupa & Stastny, 2006*; *Neustupa, 2016*; *Neustupa, 2017*). The present study primarily focused on the among-species bilaterally asymmetric variation within the lateral lobes. Each of the 1,425 analyzed cells was represented by a single lateral lobe, and thus, the intracellular variation among these structures was not evaluated. However, each mature cell includes four lateral lobes (two in each semicell). Therefore, a comprehensive analysis of asymmetry among the sublobes, such as the ULS and LLS, could include an assessment of their variation within the semicells, among the semicells within the cells, and among the cells within populations. Individual components of symmetry and asymmetry, including positional asymmetry and individual deviations from asymmetric means, could then be partitioned into these hierarchical levels of variation. However, it should be noted that the acquisition of sufficiently large datasets with mature cells represented by four fully developed lateral lobes, especially for a study investigating multiple species, would not be an easy task. Many *Micrasterias* taxa, such as *M. decemdentata*, *M. fimbriata*, *M. furcata*, and *M. jenneri*, are rather rare in natural habitats, and acquisition of sufficiently rich natural populations would not be straightforward.

The present study focused on the genetically fixed patterns of asymmetry among the species and, thus, primarily compared the amounts and differences in asymmetric means (i.e., positional asymmetry) of the studied structures. Likewise, phylogenetic morphological integration among species has been used to compare species means rather than variation within individual taxa (*Adams & Felice, 2014*). Thus, we did not analyze the patterns and amounts of FA within and among the studied populations. However, we can see that these individual asymmetric deviations from the observed positional asymmetry were significant against the digitization error in almost all of the Procrustes ANOVA runs, except for the ULS analysis in *M. denticulata*, which had unusualy high digitization error (Table S2). In contrast, high FA in shape of *M. brachyptera* sublobes yielded insignificant test statistic for the differences among the individuals (Table S2).

Amounts and patterns of FA have been linked to developmental instability (DI), leading to changes in random asymmetric fluctuations of structures (*Palmer & Strobeck, 2003*; *Graham et al., 2010*; *Klingenberg, 2015*). Although there may be multiple sources of shape FA in different model systems, a number of studies have shown that DI often increases due to various external factors, such as organic pollution, heavy metals, or suboptimal temperatures (*Klingenberg, 2015*). Interestingly, *Micrasterias* cells have recently proved to be sensitive indicators for a number of important stressors, such as heavy metals or increased environmental salinity (*Affenzeller et al., 2009*; *Volland et al., 2011*; *Andosch et al., 2012*; *Volland et al., 2014*). The heavy metals released to the environment by various anthropogenic activities may often become highly soluble in the acidic conditions of peatland pools, and thus, these metals can pose a real threat to microcommunities inhabiting these habitats. Therefore, geometric morphometric analyses of *Micrasterias* populations, focusing on FA in shape of the adjacent sublobes might be an intriguing indicator for this kind of environmental stress.

## CONCLUSIONS

This study illustrated probably the most notable morphological feature that closely relates to the phylogenetic history of the *Micrasterias* lineage. While most previously known morphological characters were poorly correlated to phylogeny, we have shown that breaking of the bilateral symmetry between adjacent lateral sublobes of *Micrasterias* cells belonging to different species is closely related to their evolutionary relationships. While members of most clades were typical by relatively compressed lower lateral sublobes and comparatively wider upper lateral sublobes, the taxa belonging to the species-rich clade "A" exhibited a phylogenetically conserved pattern with both sublobes characterized by closely similar shapes and only slight levels of positional asymmetry. Interestingly, asymmetric shape patterns between adjacent terminal lobules were largely unrelated to those of the sublobes. Thus, different hierarchical levels of cellular morphology were only weakly coordinated with regard to asymmetric variation among species. This modularity is probably the key to high evolvability of desmid shapes, leading to their extraordinary phenotypic diversity, characterized by more than 2,300 extant morphologically different species.

## ACKNOWLEDGEMENTS

The authors thank Wiley Editing Services for English language editing and style corrections.

### Funding

The study was supported by the project ''Progres no. Q43'' at Charles University, Prague. The funders had no role in study design, data collection and analysis, decision to publish, or preparation of the manuscript.

### Grant Disclosures

The following grant information was disclosed by the authors:
Charles University.

### Competing Interests

The authors declare there are no competing interests.

### Author Contributions

- Jiri Neustupa conceived and designed the experiments, performed the experiments, analyzed the data, contributed reagents/materials/analysis tools, prepared figures and/or tables, authored or reviewed drafts of the paper, approved the final draft.
- Jan Stastny performed the experiments, contributed reagents/materials/analysis tools, prepared figures and/or tables, approved the final draft.

### Data Availability

Jiri Neustupa. 2018. Data on morphometric and phylogenetic variation of 19 Micrasterias species. Knowledge Network for Biocomplexity. doi: 10.5063/F1GF0RQS.

### Supplemental Information

Supplemental information for this article can be found online at http://dx.doi.org/10.7717/peerj.6098#supplemental-information.

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
