# Peer review of "Symmetry breaking of the cellular lobes closely relates to phylogenetic structure within green microalgae of the Micrasterias lineage (Zygnematophyceae)"

_PeerJ, doi:10.7717/peerj.6098_

## Round 0.1 · original submission · Minor Revisions

The three reviewers agree that this manuscript is interesting and makes a worthwhile contribution. They all make suggestions to clarify some issues and to briefly discuss missing aspects. The manuscript would also benefit from simplifying and streamlining the text. Please consider and address these comments carefully in a revised manuscript.

·

Basic reporting

Good figures, attention to detail, Methods and Results descriptive and clear, thoughtful discussion

Language style issues: more clarity needed (specialized subject with a lot of jargon), lengthy writing style (Intro, 2nd half of discussion)

Experimental design

no issues raised

Validity of the findings

the findings and conclusions appear to be robust

Additional comments

SUMMARY
This interesting manuscript attempts to integrate morphological and molecular data in a model genus of desmids, Micrasterias. I am not an expert on this group or kind of analysis so I cannot fully evaluate the precedent evidence and the statistical methods. However, the questions are laid out and answered clearly and the figures are well presented. The analyses appear to be conclusive, the findings novel and the discussion is thoughtful, all making the manuscript worth publishing.

One thing that I was puzzled about was the use of maximum parsimony analysis for sequences: this was described in Methods but omitted in Results (or the authors did not make it clear which phylogeny they followed; I assumed ML). I wonder if MP analysis of molecular data is at all meaningful at this point (we know sequences evolve differently).

The other area in need of improvement is the writing style. The English is good but plentiful jargon makes the work difficult to understand for anyone outside the specific field. In large part, the technical terms cannot be avoided but, unfortunately, the difficulty in reading this particular text is compounded by lengthy writing, repetition and the presence of marginally relevant information. The authors should see a great space for improvement here for their own benefit. Many sentences could be more simple and direct: Introduction could be shorter by about 1/3 in size (if not more), the second half of discussion could be more focused on the important.

WRITING
Take the first paragraph of the Introduction as an example:

lines 64-65: "which is a pattern that is quite different from that of" ... "unlike". But note that you do not need this sentence.

lines 65-66: ", which often have ..") ... this is certainly not needed

lines 66-69: "Given this apparent relationship among cellular morphologies and species-level phylogenetic differentiation, it is surprising that individual major infrageneric clades .." ... "In contrast, clades of species as resolved by molecular phylogenies .." (and delete "of their member species")

To an outsider the language in the very first paragraph is rather difficult to understand. Apart from the cumbersome phrasing, I got most confused by "taxa" on line 64 (you really mean "species", I assume, but why not say "individuals" or "cells" instead?) and the fact that infrageneric clades of species are defined by the sequence phylogeny. The latter fact is important but not intuitively understood. Many similar objections to style hold for the remainder of the manuscript - aim for brevity and clarity wherever possible.

70-78: lengthy/repetitive with the first paragraph and within itself

79-84: "Neither X, emphasized in traditional taxonomy (REFs), nor Y characteristics correlate with the molecular phylogeny of Micrasterias (REFs)"

86: name the "entire lineage"

90-98: too long and some unneeded emphasis ("remarkable"). Could be replaced one or two simple sentences

99-103: "in essence", "in addition": these are parasitic words and you could write clearer. Consider this: "Micrasterias is bilaterally symmetric around multiple axes running through surface incisions; these separate the cell into cellular halves (semmicells), semmicels into lateral lobes and lateral lobes into successively branched lobules (Fig. 1A)."

105-through the end of Introduction: this contained areas with quite a lot of relevant information and the specific questions towards the end are really helpful, however, I also found it quite repetitive, unfocused (irrelevant detail) and difficult to read through (language, long sentences): the authors should cut here and simplify to be understood by readers outside their field.

Methods: could be more concise again but descriptive and generally well done

388-89: "the amounts of positional asymmetry were considerably decreased" .. awkward

404: your use of "conversely" confuses me: how does it reverse the previous statement in this case (and elsewhere)? It is not a synonym of "in contrast" if that was your intention - same issue elsewhere.

Consider similar edits across the whole manuscript

I could not find the legend for Fig. S1.

Reviewer 2 ·

Basic reporting

no comment

Experimental design

no comment

Validity of the findings

no comment

Additional comments

Review:
This manuscript focuses on an interesting topic that deals with the usefulness and power of geometric morphometrics for symmetry/asymmetry studies of unicellular algae in a phylogenetic context. In general, the manuscript is quite short, appropriately illustrated and straightforward to read. The hypotheses are clearly presented and the methods seem appropriate for the study.
I suggest adding a short definition of symmetry breaking and explain what it means and why it is relevant in this precise context. To my knowledge, the idea of symmetry-breaking (in the context of biological symmetry, which is quite distinct from the general use of the term in Physics) is devoted to the cellular processes underlying the appearance of asymmetry.

The Procrustes ANOVAs yield significant results for the main factor “individual” and the interaction term “individualxposition” for most of species. Except for the interaction term “individualxposition” in “Micrasterias denticulata, 7LM, upper sublobe” and the main factor “individual” in “Micrasterias brachyptera, 7LM, two sublobes”. In the former could it be that measurement error was rather large compared to this asymmetry measure and in the latter, could it be that this asymmetry is indeed expected to be larger than variation among individuals in this species? Maybe one comment could be added on these results.

*Main text
- line 261: “compared in two analyses (Fig. 1A).” – simply add “Fig. 1A”
- lines 316-317: Indicate which regressions were performed
- line 531: “extraordinary”
- line 537: “actually”
- line 580: “for this kind of environmental stress” – remove “the”

*Figure 2
- Somehow all images are not at the same resolution and appear as quite low resolution. Especially F), J), O) , and T).

In my opinion, the manuscript is suitable for publication after these minor corrections have been addressed.

·

Basic reporting

The present ms Neustupa & Stastsny touches a very interesting topic, mapping the phylogeny of the model alga Micrasterias to morphological traits. By a detailed analysis of symmetry breaking the authors gain important new insights. The ms is overall very well written, rich in sophisticated details and statistical treatment of the observations and is therefore well worth publishing in PeerJ.

Experimental design

In earlier work (Škaloud et al. 2011) traits like the branching pattern, complexity, cell length and even biogeographic origin were mapped on phylogenetic tree, giving important information on evolutionary trends in this interesting organism. I am wondering only about the selection of strains, why these particular species have been chosen; please mention this more precisely in the introduction; and if the rational was only what was easily accessible and growing well, this is also OK.

Validity of the findings

According to Škaloud et al. 2011 H clade Micrasterias are represented in the present study only by M. americana. It has been shown, that M. americana has very similar morphological characteristics than A-clade Micrasteris. This might be true for the here investigated lateral lobes; but to my understanding the H-clade Micrasterias show a very distinct common feature, several lobules (particularly of the polar lobe) show in addition to their 2-D architecture a 3-D. So there is not only the numbers and symmetry of the lobules in a two-dimensional view, but also the 3-D architecture well worth to be considered. This is certainly beyond the scope of the present ms, but could at least be discussed. In his post-hum article on Micrasterias, Lionel G. Harrison (Harrison, LG, 2011 Micrasterias, and computing patterning along with growth. Chapter 5, pp. 77-109. In Harrison LG, The shaping of Life. The Generation of Biological Pattern, Cambridge University Press).
This is particularly obvious in the (non-European) M. muricata where even a 3-D branching of the lateral lobes is evident (e.g. Holzinger 2000, Nova Hedwigia 70, 275-288). In future studies particularly these H-clade Micrasterias with their 3D structure should be investigated; for the present study it is enough to include this aspect in the discussion (l. 471-477) Think the ms benefits when these aspects are mentioned and the references included.
It is clear that this ms deals primarily with a mathematical and statistically sound analysis of the asymmetry between the lobules. Some aspects how this shape is realized are mentioned in the introduction, while others are missing. The membrane recognition theory of the late Prof. Kiermayer is certainly the basis for deposition of primary cell wall material, but then the quality of the wall itself is crucial for further development. I would certainly suggest to mention the important works of Vannerum et al. (2010 J. Phycol. 46, 839–845, 2011 BMC Plant Biol. 11:128), showing the responsibility of expansins in this process.

Additional comments

Other than that, this ms is smart, interesting to read, in parts certainly only for a special audience. It contributes nicely to our understanding of evolutionary trends, when investigating a genus that has so delicate morphological features.

---

## Round 0.2 · accepted · Accept

I confirm that the authors have adequately addressed the issues raised by the three reviewers.

#